# Is intergenerational elasticity (IGE) a misleading measure of wealth mobility?

**Seorin Kim**[1,2]*, **Arne Vanhoyweghen**[1], **Wouter Ryckbosch**[2], **Vincent Ginis**[1]

**1** Data Analytics Laboratory, Vrije Universiteit Brussel, Brussels, Belgium, **2** Social History of Capitalism, Vrije Universiteit Brussel, Brussels, Belgium

* seorin.kim@vub.be

**Data availability statement:** The code for the simulation and the experiment are available at https://doi.org/10.5281/zenodo.14800811

## Abstract

Intergenerational elasticity (IGE) is a widely used measure of wealth mobility, represented as the slope in an AR(1) model. While intended to capture the extent of wealth mobility between generations, this paper identifies two key issues with its use. First, the IGE provides meaningful insights only when paired with the model's convergence value, which is embedded in the intercept. A low IGE, often interpreted as high wealth mobility, does not necessarily imply that every subgroup of the population regresses to the same wealth level. Instead, it reflects the average rate at which the population converges toward the overall mean. Second, a comprehensive understanding of society's wealth mobility requires a low variance of each parameter across subgroups. A high variance suggests that different subgroups converge to different wealth levels or at different rates. In order to use the IGE as a comparative measure across countries and time periods, we suggest examining both parameters and their variance. This way, a more nuanced and thorough assessment of intergenerational wealth mobility can be achieved.

## Introduction

Intergenerational wealth mobility studies how individuals' or families' wealth position changes with respect to their parents or previous generations. If an individual's wealth is highly dependent on their parents' wealth, society's intergenerational wealth mobility is considered low, i.e., there is a lack of opportunity for the next generation to climb the social ladder. One popular measure of intergenerational wealth mobility in society is intergenerational elasticity (IGE). IGE is estimated by regressing the children's wealth against their parents' wealth, or the current generation's wealth against the previous generation's wealth. Instead of using wealth directly, the logarithm of wealth is commonly employed to ensure the normal distribution of residuals and adjust the skewness of the wealth distribution. Conventionally, a low IGE indicates high wealth mobility within society and vice versa [1].

Most studies calculate IGE using data from two generations and compare these values with IGEs from other periods or countries [2]. Despite being widely used, several issues have been identified with this approach. Firstly, employing log-transformed wealth, which necessitates non-zero wealth values, can be problematic depending on how wealth is defined and what age range is considered for the younger generation [3]. The use of log-transformed wealth is further criticized by [4] as it does not allow IGE to be interpreted as a regression to the

**Funding:** VG acknowledges funding (IBOF-23078) from Vrije Universiteit Brussel (vub.be). The funder did not play any role in the study design, data collection and analysis, decision to publish, or preparation of the manuscript.

**Competing interests:** The authors have declared that no competing interests exist.

arithmetic mean but a geometric mean. These criticisms around the use of log-transformed wealth have favored other measures, such as the rank-rank coefficients (e.g., [5]), which assess the link between a previous generation's rank in wealth and a current generation thereof [3].

Another notable issue with IGE is attenuation bias, where the estimated value of IGE is systematically lower than it should be [6–8]. [6] explains that this bias arises from transitory fluctuations, similar to life-cycle bias [8], and/or homogenous samples. Lastly, [9] pointed out the limitation of IGE in cross-country comparisons by investigating the model through Yitzhaki's theorem, which interprets the regression coefficient of an OLS regression "as a weighted average of [the regression coefficients] defined by adjacent observations in the sample" [10, p.1].

In addition to the previously discovered issues regarding IGE, we argue that the conventional approach lacks comparability between groups across countries and time periods on the one hand and representativity when some subgroups' wealth evolves significantly differently from others on the other hand. As will be further explained throughout the paper, the first issue derives from the fact that one of the parameters is ignored, and the second issue can be explored by looking at the variances of the parameters. Therefore, we highlight the importance of considering another parameter—namely, the convergence value in addition to the IGE value—as well as the variances of the two parameters.

We will first outline the primary issues with the current use of IGE. For this, we focus on the fact that the IGE model implies a first-order autoregressive or AR(1) model [2], which assumes time-series data. Then, we explore each issue in detail across two sections. Considering the attractiveness of using IGE for its simplicity and convenience, we suggest the way in which its representativity and comparability across time periods or countries can increase rather than proposing a new measure. This will be done in the fourth section, illustrating the theory with the probation data from [11]. In the subsequent sections, we will refer to the IGE parameter as an autoregressive coefficient, which is the regression coefficient in an AR(1) model or simply $\beta$ in Eq (1).

## The IGE model

To briefly illustrate the problem, consider the two cases in Fig 1 representing the log wealth dynamics of families and the population's expected log wealth over time over 20 generations. Two family types are considered: those with a high starting wealth with a mean starting log wealth of 10 in gray and those with a low starting wealth in the first generation with a mean of -10 in red. Interestingly, despite the large difference in the IGE values, both cases demonstrate the mixing of both types of initial wealth over time, albeit individual trajectories converge differently. If one believes that a highly mobile society coincides with wealth redistribution over the population, Case (B) is preferred to Case (A) since both types of families gradually converge to the population mean. However, conventionally speaking, the lower IGE of 0.3 in Case (A) is typically interpreted as the society having higher wealth mobility than in Case (B).

Comparing these two cases clearly highlights that the IGE value is not the only parameter determining whether individuals' wealth can grow independently of what their parents had. Instead, the convergence value, which determines the model's intercept (see S1 Appendix), influences individuals' and society's wealth redistribution over time. Moreover, it is remarkable that the population-wise convergence value does not always represent the individuals' or subgroups' convergence values. In other words, the value to which the population's wealth converges, how fast it is going, and whether everyone is evolving toward the same

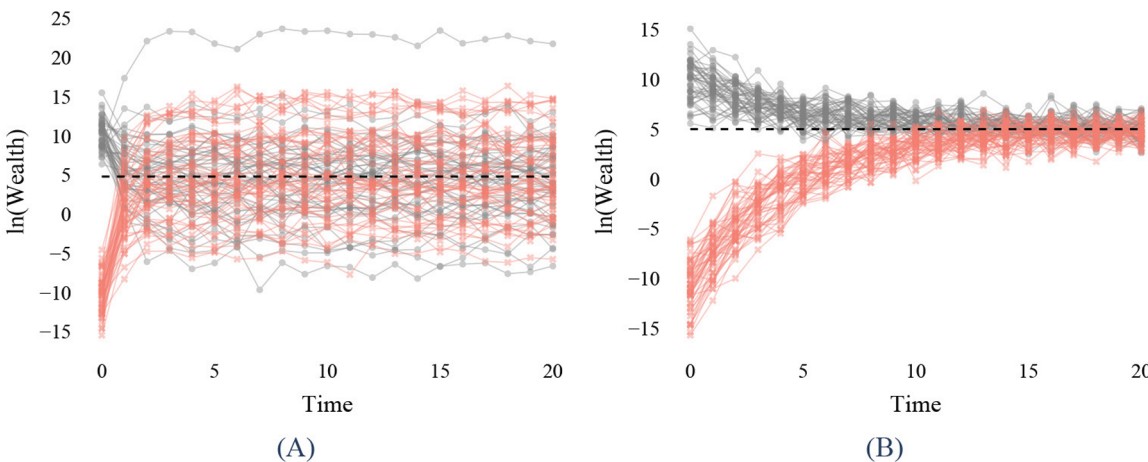

**Fig 1. Paradoxical relationship between IGE and wealth dynamics.** Two cases are presented where the log wealth of two groups with contrasting initial wealth evolve over 20-time steps/generations. In (**A**), the convergence values of the population and the subgroups are different, while they are the same in (**B**). The population-wise IGE value is 0.3 in (A) with a convergence value of 5, and the IGE value of 0.8 with a convergence value of 5 in (B). The subgroup-specific wealth dynamics are gray for those with high initial wealth (●) and red (✖) for those with low initial wealth, and the population-wise convergence value is the black dashed line. The lower IGE in (A) is preferred to (B) in a conventional interpretation. However, the graphs demonstrate that (B) redistributes wealth across populations of different initial wealth more than (A).

convergence value at the same speed are key to understanding the wealth mobility of society.

The next two sections will demonstrate the importance of, first, considering the two parameters in the IGE model, i.e., the convergence value, $\mu$, and the IGE value, $\beta$, when discussing wealth mobility and, second, taking into account the variance of each parameter.

For a discrete-time $t \geq 1$, the conventional model for IGE is given by

$$\ln W_{it} = \alpha + \beta \ln W_{it-1} + \varepsilon_{it}, \tag{1}$$

where the intercept $\alpha = \mu(1 - \beta)$, $W$ represents wealth, $i$ indicates a family or individual, and $t$ is a generation. The white noise $\varepsilon$ is assumed to be normally distributed with a mean of zero and a variance, $\sigma^2$. In order to converge, the value of $\beta$ should lie between 0 and 1 (shown in S1 Appendix). Notice that the structure of this model implies an AR(1) model: the log wealth of a previous generation or a time step $t-1$ influences the log wealth of the current generation or $t$. While this structure allows $\ln W_{it-1}$ to be independent from $\varepsilon_{it}$, $\ln W_{it-1}$ is not independent of $\varepsilon_{it-1}$.

## The role of the two parameters of the IGE model: $\mu$ and $\beta$

Unlike the conventional use of the model, we argue that both the convergence value, $\mu$, and the autoregressive coefficient, $\beta$, are important to understand a society's wealth mobility and further allow some comparisons across countries or periods of the same society. We will demonstrate the importance of investigating the two parameters by discussing each role in the model.

Firstly, the convergence value, $\mu$, determines where the population's log wealth evolves over time. Fig 2a demonstrates how the log wealth dynamics can differ with convergence

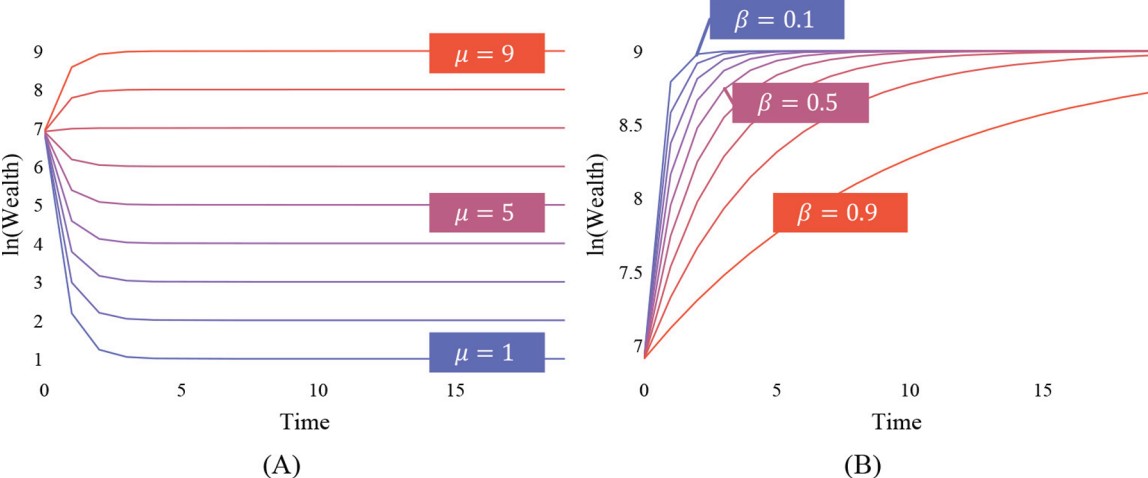

**Fig 2. The role of $\mu$ and $\beta$.** The X-axis represents time or generations, and the Y-axis represents log wealth. (**A**) demonstrates the role of $\mu$ through 9 cases with varying $\mu$ from 1 in blue to 9 in red. All cases have the same low $\beta$ of 0.2 and the same initial wealth of 1000. (**B**) demonstrates the role of $\beta$ by varying $\beta$ from 0.1 in blue to 0.9 in red. All cases have the same $\mu$ of 9 and the same initial wealth of 1000.

values from 1 to 9 when $\beta$ is fixed at 0.2. Despite the equally low $\beta$, we can learn that some societies may have their log wealth tending to a higher or lower value than their initial wealth, depending on the $\mu$ given to the society. Therefore, considering both the coefficient and the convergence value is crucial to establish a foundation for comparing the wealth mobility of different societies. As will be discussed in the later section, this point is still relevant when examining one society—namely, in that the convergence value may differ among different subpopulations within a society.

Secondly, the coefficient, $\beta$, determines how fast the log wealth converges to the population mean. It should be noted that the log wealth converges to $\mu$ only when $\beta$ lies between 0 and 1. Given the equal convergence value and the initial wealth, the $\beta$ of 0.1 in Fig 2b moves to the convergence value more quickly than the $\beta$ of 0.9. Therefore, the coefficient determines how fast log wealth converges to $\mu$ as a smaller $\beta$ requires fewer generations $c$ such that the $\lim_{t \to c} e^{(1-\beta^t)\mu} W_{i0}^{\beta^t} = e^{\mu}$.

In addition to $\beta$, the convergence speed is also influenced by $\mu$. The larger $\mu$ in Fig 2a allows the log wealth to converge slightly quicker to its population mean than the smaller $\mu$. This can be numerically derived by solving $W_{it} = e^{\mu + \epsilon}$ for a small $\epsilon$ which is the difference between a log wealth value at time $t$, $W_t$, and the convergence value, $\mu$. This leads to $t = \ln\left(\left|\frac{\epsilon}{\ln W_{i0} - \mu}\right|\right) \cdot \frac{1}{\ln \beta}$ and tells us $t$ generations will be spent to get very close as $\epsilon$ to $\mu$. Although this effect of $\mu$ on the convergence speed is smaller than $\beta$, it reveals that $\beta$ is not the only source of the convergence speed.

Considering the role of these parameters, it becomes clear that solely reporting the coefficient only allows us to know how fast the population converges to the population mean. Therefore, knowing the convergence value is crucial for achieving a fair comparison between societies or time periods. In the next section, we will further see the importance of inspecting the variances of these parameters among subgroups of the population.

## Unveiling the importance of the variances of $\mu$ and $\beta$

While reporting IGE allows us to know the convergence speed and how much the previous generation's wealth resonates with the current one's, it should be paired with its variance across subgroups. As for the convergence value, including its variance across subgroups broadens the view of a society's intergenerational wealth mobility. In this section, the variances of $\mu$ and $\beta$ mean the variance in the distribution of, respectively, subgroup-specific $\mu_i$ and $\beta_i$ for $i$ subgroups. We consider 100 subgroups in the experiments—hence, $i$ is set to $1, \ldots, 100$.

Consider the four cases in Fig 3 where the two parameters of two different variances are illustrated. Firstly, we recognize that when both parameters' variances are small (A), the log wealth of most subgroups regresses to the population mean. Thus, wealth is redistributed across the population, and the subgroups of different initial wealth eventually converge to the population mean at a similar speed. When both parameters vary widely across subgroups (D), each converges to a different value at a varying speed. In this case, the estimate of each parameter does not represent the subgroups' wealth dynamics. When $\mu$ is narrowly but $\beta$ is

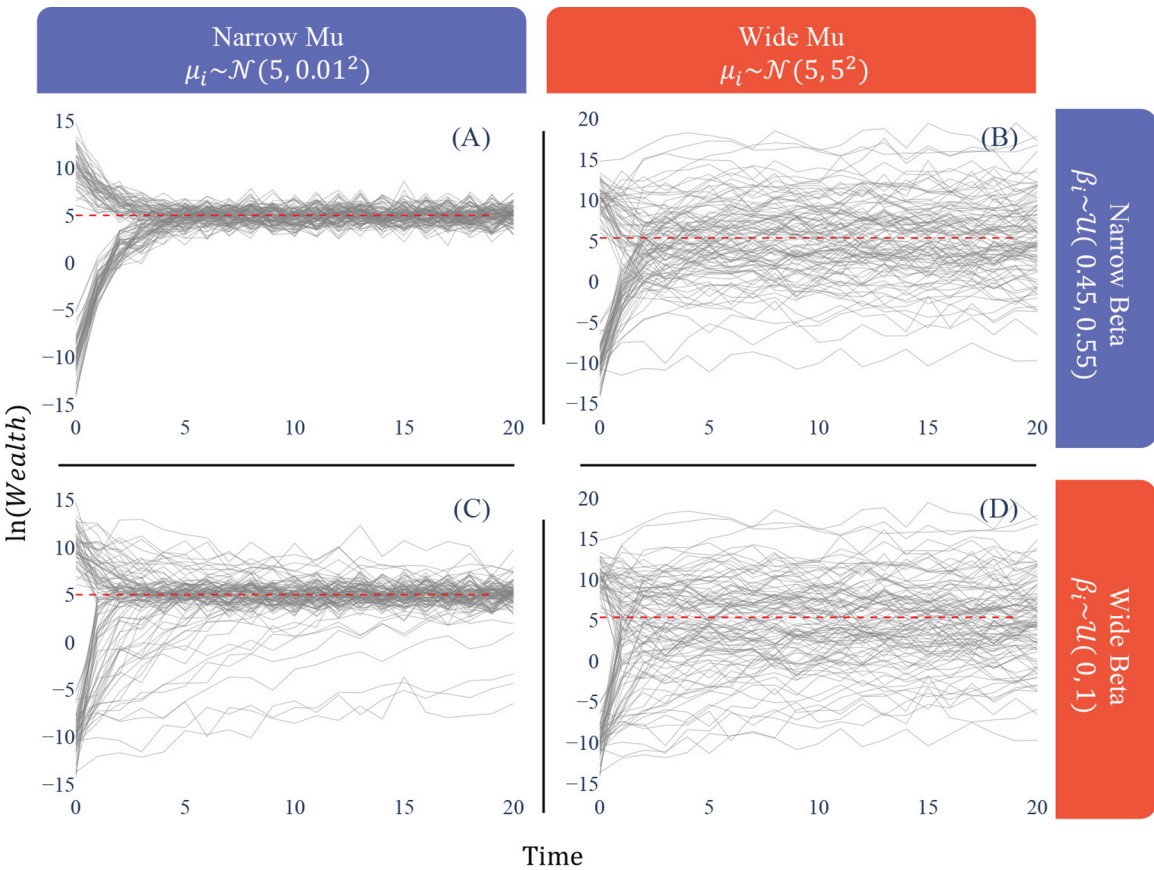

**Fig 3. The relationship between variances of subgroup-wise $\mu$ and $\beta$.** The subgroups' $\mu_i$ are randomly given from a normal distribution, and their IGE values are from a uniform distribution. (**A**) both parameters with small variances, $var(\mu) = 0.001$, $var(\beta) = 0.0008$; (**B**) a large $var(\mu) = 25$ but a small $var(\beta) = 0.0008$; (**C**) a small $var(\mu) = 0.001$ but a large $var(\beta) = 0.0833$; (**D**) both large $var(\mu) = 25$, $var(\beta) = 0.0833$. Assuring a small variance of both parameters—especially $\mu$—appears crucial to achieving wealth redistribution of different wealth levels.

widely distributed (C), subgroups eventually converge to the population mean, but some take significantly longer than others to reach that point. Lastly, when $\mu$ varies widely between subgroups (B), but the variance of $\beta$ is small, every subgroup moves towards its own mean at a similar speed. In this case, the subgroups of varying initial wealth take a long time to regress to the population mean. These examples highlight the importance of inspecting the variance of each parameter for the representativity of their estimates. Especially, a small variance of $\mu$ seems crucial if one aims to achieve a system where wealth is redistributed over different quartiles of the population.

We also consider the impact of the bimodality of the parameters, which is another form of a large variance, on wealth redistribution. For this, two sets of simulations are run. First, we compare a case where the subgroups' $\mu_i$ values are around either of two modes to a case where $\mu_i$ are distributed with a small variance around one mode across subgroups. Second, we compare a case where the distribution of the subgroups' $\beta_i$ values is bimodal to a case where it is unimodal around 0. In all cases, the bimodality is achieved with a large between-mode variance and a small within-mode variance. Moreover, the same expected $\beta$ and equally fixed $\mu$ are given to the unimodal- and bimodal $\beta$ cases (i.e., $\hat{\beta} = 0.45$ and $\mu_i = 0$ for $i = 1, \dots, 100$), and the same expected $\mu$ and equally fixed $\beta$ to the two $\mu$ cases (i.e., $\hat{\mu} = 0$ and $\beta_i = 0.45$ for $i = 1, \dots, 100$).

Again, Fig 4 depicts that the impact of a small variance of the convergence values appears to be the strongest in achieving wealth redistribution under the equally fixed $\beta$ for all individuals. The bimodal $\beta$ values result in having one group converging to the population mean later than the other.

Kendall rank correlation coefficient (hereafter Kendall's $\tau$) allows us to understand how much of the blending of subgroups with varying initial wealth happens at each time step compared to the initial state. Kendall's $\tau$ compares the wealth ranking at a time step $k$ to time step 0. As $\tau$ ranges between 1, meaning all the rankings of items in the compared sets concord, and -1, being all are discordant, 0 alludes that the wealth ranking at a time, $t_k$, is independent of the initial wealth ranking [12].

When comparing the small and large variances of the parameters, (A) and (B) in Fig 5, we first learn that the narrow $\beta$ ensure a quicker convergence of Kendall's $\tau$ to 0 than the wide $\beta$ when both have a narrow $\mu$. However, with an equally narrow $\beta$, the wide $\mu$ seems to achieve the zero state quicker than the narrow $\mu$. When comparing different modalities of the parameters, (C) and (D) in Fig 5, the resonance of the initial wealth stays longer in the bimodal $\beta$ case than in the unimodal $\beta$ case. However, getting rid of the influence of the initial wealth becomes challenging when the $\mu$ given to a group is significantly different from the other group. As (C) shows, the $\tau$ value in the bimodal $\mu$ case converges around 0.5 while that in the unimodal $\mu$ converges toward 0.

Taken together, unimodality is crucial for the convergence value, $\mu$, ensuring wealth redistribution. As long as the unimodality is guaranteed, the ranking of wealth will change over time. However, the IGE value, $\beta$, requires both a small variance and unimodality to not stray from conventional interpretations of IGE. Moreover, understanding the comparison between the wide and narrow $\mu$ cases, (A) and (B) in Fig 5, with Fig 3 illuminates that Kendall's $\tau$ does not necessarily indicate the variance of the mixing of the population. In other words, the low $\tau$ may indicate that wealth is redistributed, but it does not assure whether everyone has a similar wealth.

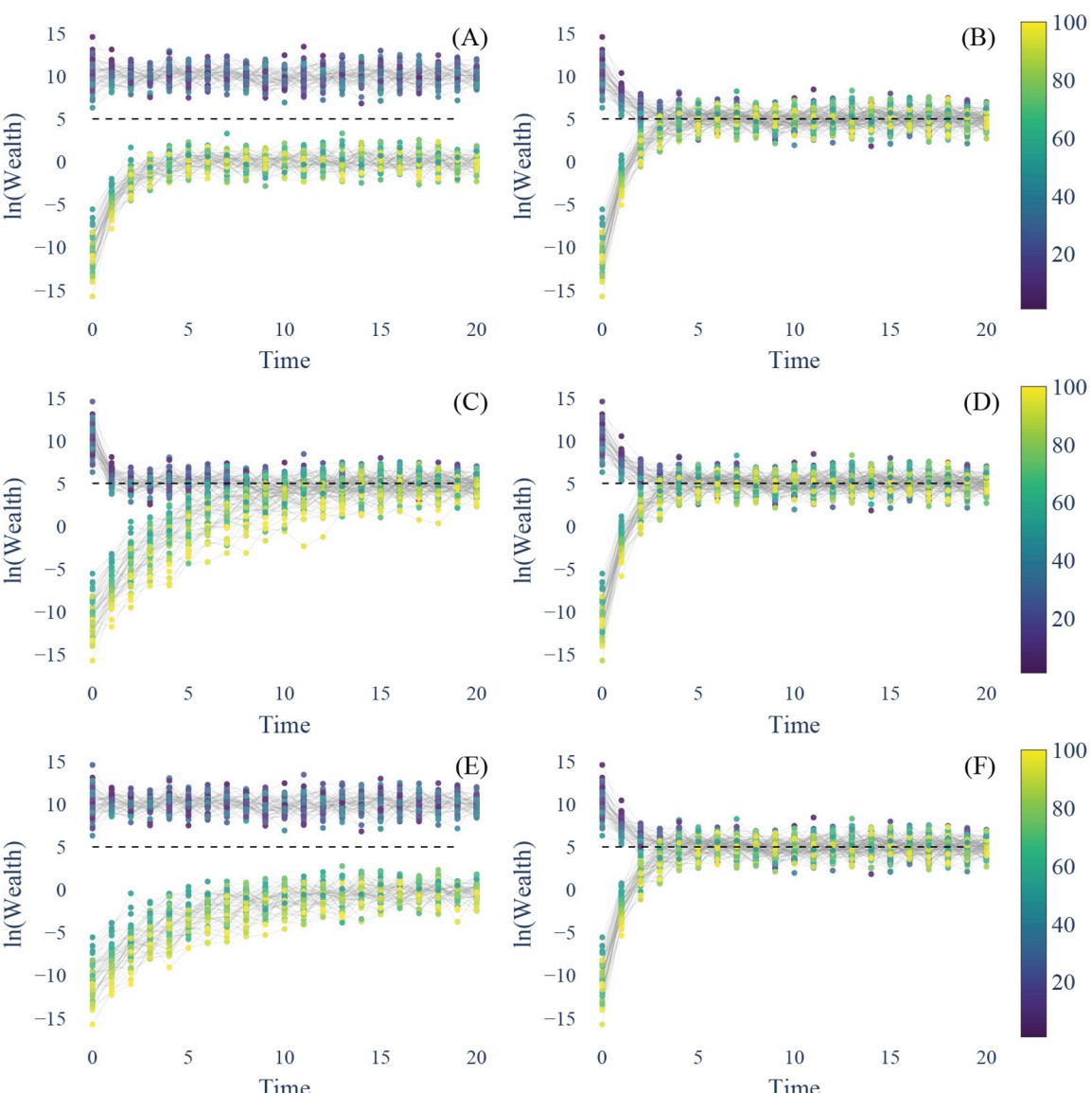

**Fig 4. The effect of uni- and bimodality of $\mu$ and $\beta$.** (A) Bimodal $\mu$ case: $\mu_i \sim \mathcal{N}(10, 0.01^2)$ for the half of the population and $\mu_i \sim \mathcal{N}(-10, 0.01^2)$ for the other half and $\beta$ is fixed to 0.45. (B) Unimodal $\mu$ case: $\mu_i \sim \mathcal{N}(5, 0.01^2)$ for all with a fixed $\beta = 0.45$. (C) Bimodal $\beta$ case: $\beta_i \sim U(0, 0.2)$ for the half of the population and $\beta_i \sim U(0.7, 0.9)$ for the other half, and $\mu$ is fixed to 5. (D) Unimodal $\beta$ case: $\beta_i \sim U(0.4, 0.5)$ for all with a fixed $\mu = 5$. (E) Both $\mu$ and $\beta$ have bimodality. (F) Both $\mu$ and $\beta$ have unimodality. The population size is 100. The expected IGE value at the population level is 0.45 for (C) and (D) and the population-wise convergence value is 5 for all.

## Experimentally illustrating the theory with the data of Clark and Cummins [11]

In this section, we showcase the discussed points—namely, the importance of recognizing the convergence value in addition to the IGE value and their variances—using the probation data from [11]. The authors offer the data in two formats: a dataset with individual probation and the other where the child's and the father's wealth at death are linked. Note that these authors

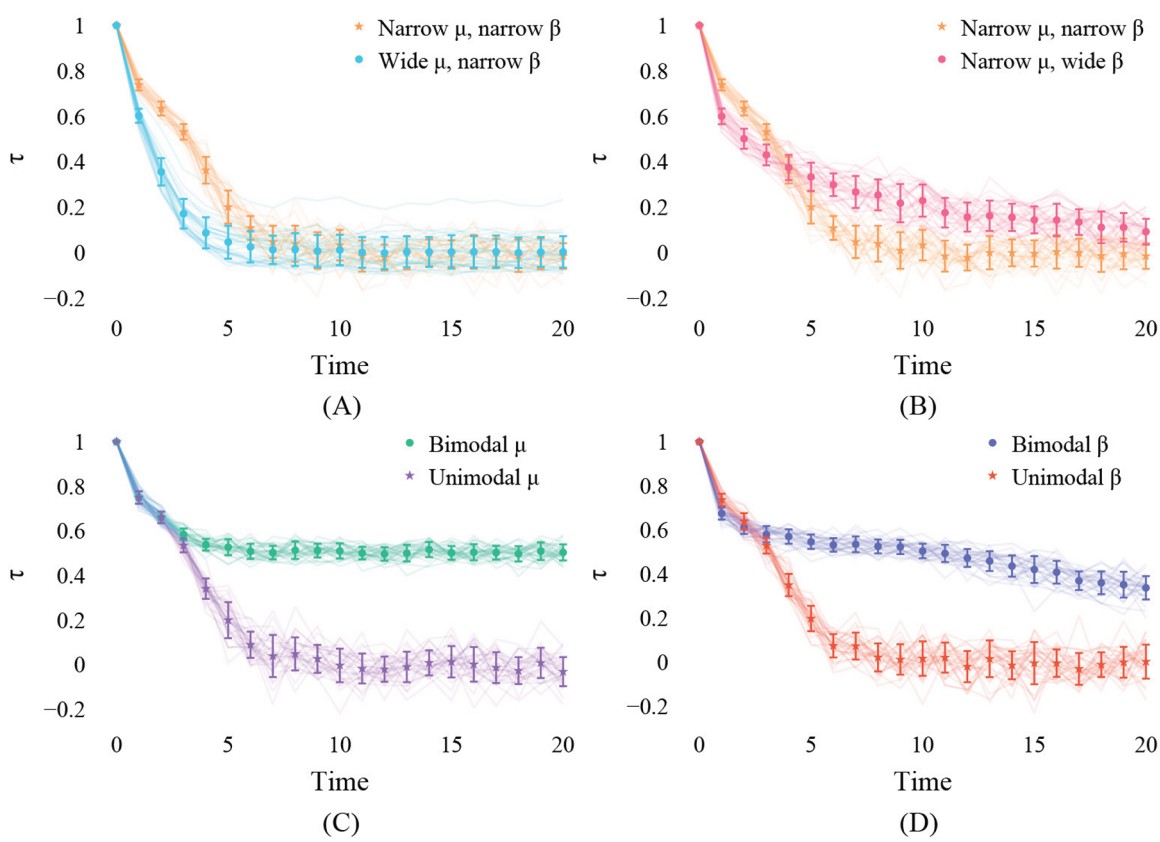

**Fig 5. Kendall's $\tau$ of the cases with different variances in $\mu$ and $\beta$.** (A) Narrow vs. wide $\mu$ both with a narrowly distributed $\beta$ (A vs. B in Fig 3); (B) Narrow vs. wide $\beta$ both with a narrowly distributed $\mu$ (A vs. C in Fig 3).(C) Bimodal $\mu$ case vs. Unimodal $\mu$ case (D) Bimodal $\beta$ case vs. Unimodal $\beta$ case. The setups are the same as in Fig 4, but 30 replications are made for each scenario. $\tau$ of 0 signifies the state in which the initial wealth's ranking no longer matters. Comparing the results to Fig 3, this reveals that $\tau$ does not directly translate into whether all the population converges to the same wealth level.

had to make some assumptions to create a longitudinal dataset. One of their assumptions is considering all the individuals of the same surname as the same family. Moreover, the authors opted for rare surnames to reduce the variability within the same surname.

We will use the individual probation dataset from [11] to explore the variances of convergence values and IGE values. The dataset includes 18869 individuals of five generations who belong to one of the 486 rare surnames considered in the study. The generations are defined by a range of the years when children died. It should be noted that not all surnames have complete data for the considered generations. In order to yield reliable estimates by surname-specific AR(1) models, only the surnames with at least 30 data points in each interval of two generations are considered. Note that this is also why we opted to work with the individual probation dataset rather than the linked one. After running an AR(1) model as in Eq (1) for each surname, the surnames whose autoregressive coefficient does not lie between 0 and 1 are disregarded, resulting in 326 surnames (hence, $i = 1, \ldots, 326$). The same set of surnames is considered to generate the population-wise estimates from one IGE model.

When considering the variance of $\mu$ at each time period in Fig 6, we observe a large variance at each time step. In the second and third periods, two modes are apparent, which may suggest the existence of a variable that groups the population into two (e.g., rich and poor

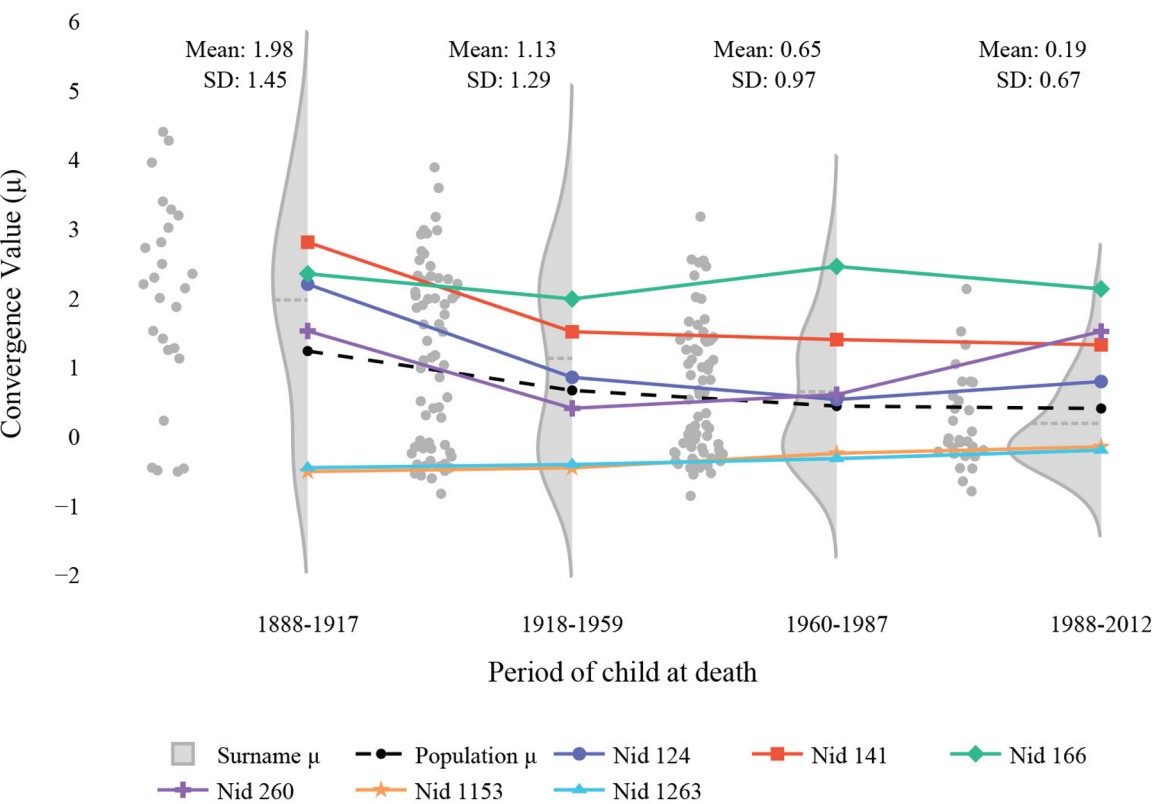

**Fig 6. The relationship between subgroup- and population-wise $\mu$ with Clark and Cummins' dataset [11].** The first period contains IGE values calculated with respect to the previous generation, 1858-87. The distributions of surname-wise $\mu_i$ are plotted at each time period with their mean values in gray dashed lines and their respective scatters on their left side. The evolution of population-wise $\hat{\mu}$ is in a black dashed line. Other colored lines depict the evolution of the surnames' $\mu_i$, whose values exist for all the periods. `Nid` refers to the surname ID. Notably, the presented surnames' $\mu_i$ evolve differently from the population's $\hat{\mu}$.

groups or different educational backgrounds). For the first and the last period's distribution of $\mu_i$, one mode is shown. However, especially in the last period, the mode does not lie with the surname's mean convergence value (the gray dashed line in the density plot), highlighting the impact of some high $\mu_i$ values. Moreover, the population-wise $\hat{\mu}$ values change over time, which makes it hard to interpret the IGE in a conventional way.

While the variability in the convergence values within each time period and between these periods may echo with the unstable wealth dynamics described in [13], it might be unclear what the next step should be. We suggest here that when such variability is observed, investigating a latent variable that leads to different wealth evolution is crucial. Moreover, if the data is traceable over time (i.e., time-series data where either an individual or a subgroup can be followed over time), one can also compare how the distributions of $\mu_i$ differ between time steps and look for the source of divergence. In Fig 6, we can trace over surnames, albeit not all surnames exist in all periods. Six surnames with complete data throughout the periods demonstrate instability in their convergence values over time. Surname ID 141, for instance, shows a gradual decrease in the convergence values, which may be due to large heterogeneity within the surname, while nid 1153 and 1263 showed relatively similar convergence values over time. Therefore, studying the difference in the individual wealth distribution of surname

ID 141 between the first and the last periods can offer a fuller picture of the wealth mobility of this society than reporting the IGE value alone.

When investigating the variance of $\beta$ at each time period in Fig 7, the population-wise IGE values (black markers) are larger than the mean of the surname-specific IGE values at each time period (gray dashed lines in density plots). This large discrepancy between the population-wise estimate and the mean of the surname-wise estimates can be surprising at first glance. However, given the inherent downward bias in the AR(1) estimator, especially with the small sample size, the smaller sample sizes in surname-wise IGE models might have been more influenced by this downward bias than in the population-wise IGE model [14–17]. Another possibility is that considerable heterogeneity may exist within the surnames—that is, an opposite case of [6] where the author demonstrated that homogenous samples systematically yield a downward bias in the population-wise estimate.

Similar to the distributions of $\mu_i$, the distributions of the surname-specific $\beta_i$ are not always Gaussian. The distributions are often skewed to the lower values with large variances, and the coefficient of variance (i.e., the ratio of the standard deviation to the coefficient) exceeds 0.6 across all periods. This suggests that surnames are not converging at the same speed at each period.

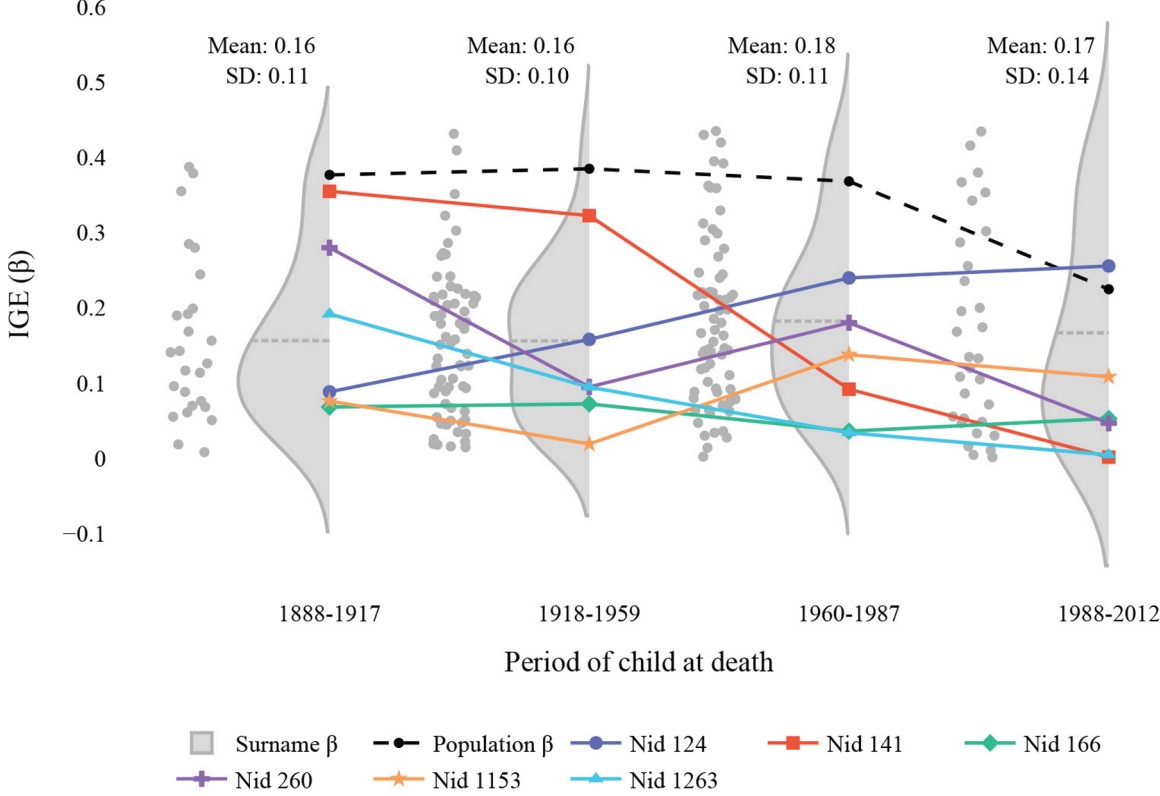

**Fig 7. The relationship between subgroup- and population-wise $\beta$ with Clark and Cummins' dataset [11].** The first period contains IGE values calculated with respect to the previous generation, 1858–87. The subgroup-specific IGE distributions are plotted at each time period with their mean IGE values in gray dashed lines and their respective scatters on their left side. The evolution of population-wise IGE is in a black dashed line. Other colored lines depict the evolution of the IGEs of the surnames whose IGE values exist for all the periods. Notably, the presented surnames' IGEs evolve differently from the population's IGEs.

Fig 7 shows that each surname that exists at all the generations experiences different evolutions of IGE from each other and the population. For instance, the IGE of surname ID 141 in a red line has a similar IGE to the population's, around 0.4 in the first period, but afterward, its value drops near 0 at the end. On the other hand, the IGE of surname ID 124 in blue started with a very low IGE of around 0.1, then skyrocketed above the population's value at the end. Therefore, without considering the variance of $\beta$, the IGE estimate overlooks the subgroup-wise differences in their convergence speed.

To further investigate society's wealth mobility, Kendall's $\tau$ on the log wealth values of individuals can be useful. For this, we use the linked dataset by [11] where the log wealth of the child and the father are given for each time period. After filtering out the missing values for the father's and child's wealth and the time period variables, 4952 observations of the original 6927 are considered in the analysis. Note that the time period here is the period when the child died, not the father.

Table 1 shows that the $\tau$ values are all below 0.4. The low but positive Kendall's $\tau$ values indicate that wealth ranked highly in the father's generation tends to retain a relatively high rank in the child's generation, suggesting a modest persistence of wealth across generations. In order to visually understand how each quartile of the population moved between the two generations at each interval of time, we created heatmaps. This is similar to the Markovian approach to social mobility (e.g., [18–21]). The heatmaps in Fig 8 further show that, still, over 50% of children in the top 20% wealth bracket have fathers who also belong to the top 20% wealth bracket in all the periods except the first period. However, for the lower wealth brackets, the proportions are smaller. Note that we use 20% wealth brackets for a coherent display of the five heatmaps, but this resulted in having a small sample size for some cells in period 1.

While Fig 8 provides insight into the proportion of wealth transmitted across generations, the transmission matrices represented by the heatmaps do not account for the possibility that fathers in the top 20% wealth bracket may themselves originate from this same bracket of the previous generation [18]. In essence, this visualization is limited in its ability to capture long-term wealth dynamics, as it focuses on static snapshots of wealth transmission between two generations without accounting for broader temporal trends or cumulative effects over multiple generations. Additionally, the observed patterns could vary when subdividing the population into finer wealth categories or incorporating other variables such as surnames [22,23]. Consequently, Fig 8 should be interpreted as complementary to the earlier analyses, offering a visual perspective on intergenerational wealth mobility rather than a standalone conclusion.

## Rank-Rank slope and accounting for the zero-wealth problem

In our experiments, non-zero wealth is assumed for all the individuals given the conventional IGE model with the log-transformed wealth. When zero-wealth (or zero earnings or

Table 1. Kendall's tau by comparing the child's wealth to their father's wealth at each period, using the linked dataset.

| Period of Child's Death | $\tau$ |
|---|---|
| 1858-1887 | 0.291 |
| 1888-1917 | 0.398 |
| 1918-1959 | 0.356 |
| 1960-1987 | 0.299 |
| 1988-2012 | 0.243 |

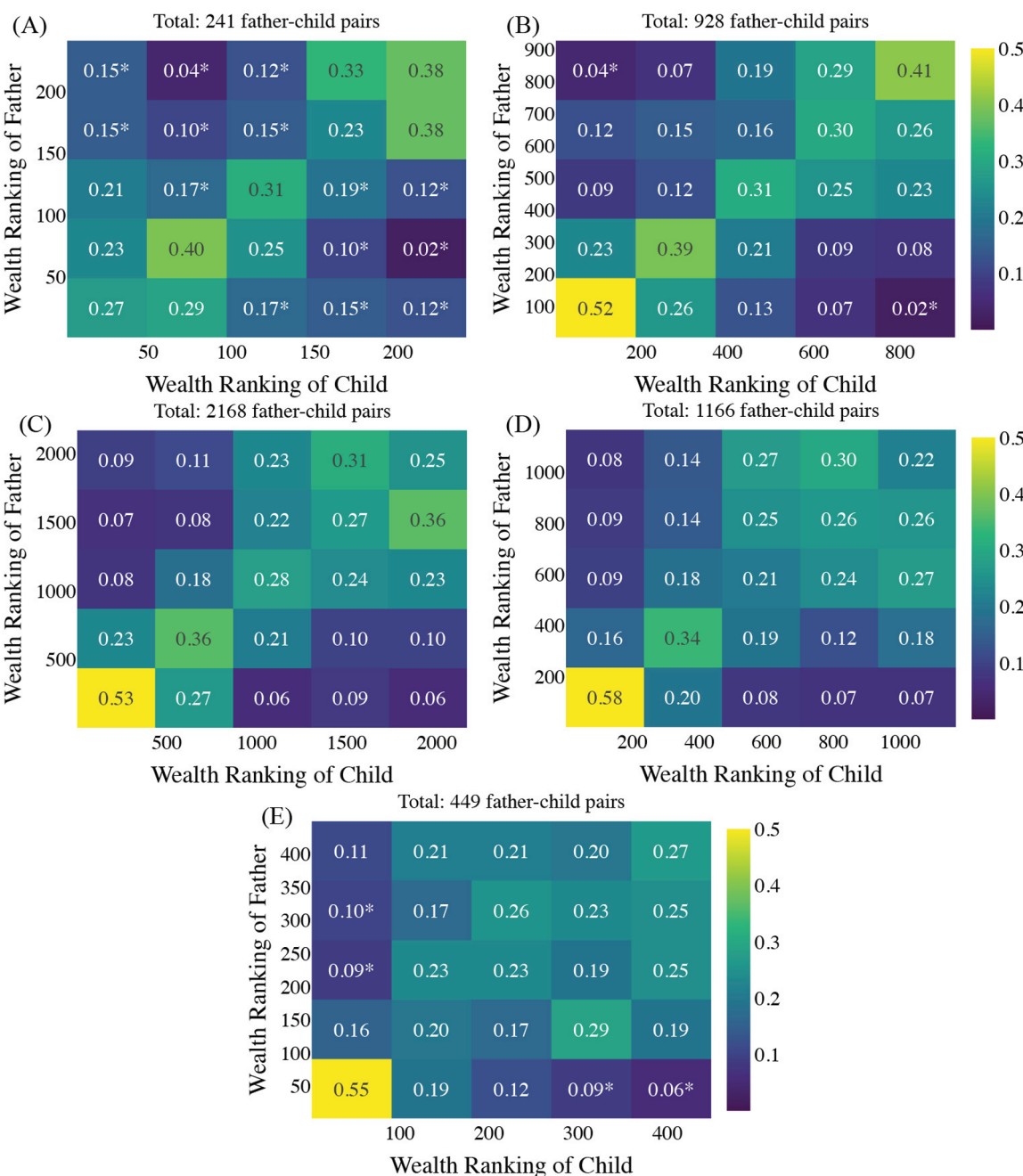

**Fig 8. Transmissions in wealth rankings between fathers and children, using the linked data.** Both wealth rankings are divided into five bins. Each cell in the heatmap shows the proportion of children (along the x-axis) whose fathers (y-axis) fall into a particular wealth ranking bin. The color scale represents the density within each x-bin, where a vertical sum for each x-bin is normalized to 1, meaning the color in each column reflects the distribution of the father's wealth ranking within that child ranking bin. Brighter (yellow) colors indicate a higher proportion of children in that wealth ranking bin whose fathers were in a corresponding wealth ranking bin. An asterisk is given to the cells where the sample size is smaller than 10.

income) is included in the data, it may be preferable to use the rank-rank correlation coefficient instead of the IGE, as [24] suggest, to address the challenges posed by zero-wealth cases. The rank-rank coefficient is given by:

$$r_{it+1} = a + br_{it} + \epsilon_{it}, \tag{2}$$

where $r$ is the ranking of the $i^{\text{th}}$ subgroup at time $t$, $a = m(1 - b)$ and $\epsilon_{it}$ is a white noise. When $|b| < 1$,

$$r_{it} = b^t r_{i0} + (1 - b^t)m, \tag{3}$$

which shows that in limit, $r_{it}$ converges to $m$. With or without the log transformation of the variable, it will be crucial to consider both the convergence value and the slope estimate and their variances.

## Conclusion

Intergenerational wealth elasticity is frequently used to examine how many opportunities individuals have for wealth growth regardless of their financial background. The same AR(1) model for wealth mobility is used to measure intergenerational elasticity in income, education, and more. For instance, [25] uses the IGE model with earnings to observe the relationship between inequality and intergenerational mobility in earnings, and the World Bank uses this model to track social mobility based on individuals' years of education [26]. However, as this paper reveals, this measure has multiple issues, which are equally important to consider when examining mobility in income, education, and occupation. In other words, these two issues are equally prevalent in another widely used measure based on an AR(1) structure— the rank-rank coefficient—regardless of the type of variable used in the model (e.g., wealth, income, or earnings).

The paper demonstrates the paradox where the conventional IGE can portray two different wealth dynamics with the same IGE value: One with wealth redistribution over every quartile of the population and the other where no redistribution happens. We argue that this paradox comes from (1) ignoring another parameter in the IGE model, i.e., the convergence value, $\mu$, and (2) overlooking the variances of the two parameters. Reporting the convergence value allows us to understand society's wealth dynamics and opens up opportunities to compare them to those of other countries or periods. Inspecting the variances of the convergence value and the IGE value ensures the representativity of these values. Especially when the convergence value is unimodal with a small variance, wealth redistribution is guaranteed across different population quartiles. Then, IGE can be further used to investigate which subgroups reach the population mean faster than the other. Even when the variances are large for both parameters, exploring the variances is still advantageous for many researchers as it can unearth a hidden structure behind wealth mobility.

While we suggested an approach to focus on both parameters and their variances, an alternative method for summarizing analysis results is not proposed. Instead, with data from [11], we demonstrated some visualizations of wealth redistribution across periods with Kendall rank correlation coefficient and heatmaps. While these can serve as additional tools to investigate the divergences within the population's wealth transmission, further research is needed to develop a practical alternative to the conventional IGE method. Nonetheless, we believe that our approach can already provide valuable insights into wealth mobility, broadening the comparability and representativity of the measure.

## Supporting information

**S1 Fig. Kendall's $\tau$ based on the log wealth data aggregated for surnames.** If one aims to observe the evolution of $\tau$ over periods, the researcher may consider a subgroup to aggregate the individual wealth. With the individual wealth dataset from [11], the wealth values are aggregated for surnames. Here, to be able to have the same length of the data at each period, we only consider the surnames that have data points for all five periods, resulting in 88 surnames over the five periods, i.e., 440 observations. Then, Kendall's tau is calculated for each period, with the first period as the reference time to compare. The log wealth values of individuals of a surname are aggregated in two ways: median and mode. The blue line connects the $\tau$ values calculated with the surname's mean log wealth values, and the red line connects the $\tau$ values calculated with the surname's median log wealth values. $\tau = 1$ at the first period since the first period's log wealth rank is the reference, and the value quickly decreases over time.
(TIFF)

**S1 Appendix. Wealth dynamics implied in the IGE model.** In the following, we show the wealth dynamics implied in the IGE model by expressing the model with respect to the initial wealth, the convergence value, and the slope.
(TEX, PDF)

## Author contributions

**Conceptualization:** Arne Vanhoyweghen, Vincent Ginis.

**Data curation:** Seorin Kim.

**Formal analysis:** Seorin Kim, Arne Vanhoyweghen.

**Funding acquisition:** Wouter Ryckbosch, Vincent Ginis.

**Investigation:** Seorin Kim.

**Methodology:** Seorin Kim, Arne Vanhoyweghen.

**Project administration:** Seorin Kim.

**Resources:** Seorin Kim.

**Software:** Seorin Kim.

**Supervision:** Wouter Ryckbosch, Vincent Ginis.

**Validation:** Seorin Kim.

**Visualization:** Seorin Kim.

**Writing – original draft:** Seorin Kim.

**Writing – review & editing:** Seorin Kim, Arne Vanhoyweghen, Wouter Ryckbosch, Vincent Ginis.

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
