## [Decision Letter · Decision Letter 0]

21 Jan 2025

PONE-D-24-56261Is Intergenerational Elasticity (IGE) a Misleading Measure of Wealth Mobility?PLOS ONE

Dear Dr. Kim,

Thank you for submitting your manuscript to PLOS ONE. After careful consideration, we feel that it has merit but does not fully meet PLOS ONE’s publication criteria as it currently stands. Therefore, we invite you to submit a revised version of the manuscript that addresses the points raised during the review process.

Based on the advice received, I have decided that your manuscript could be reconsidered for publication should you be prepared to incorporate major revisions. When preparing your revised manuscript, you are asked to carefully consider the reviewers comments which can be found below, and submit a list of responses to the comments. You are kindly requested to also check the website for possible reviewer attachment(s).

We look forward to receiving your revised manuscript.

Kind regards,

Marco Maria Sorge, PhD

Academic Editor

PLOS ONE

Journal Requirements:

3. We notice that your [Supplementary-material pone.0324266.s002]. are included in the manuscript file. Please remove them and upload them with the file type 'Supporting Information'. Please ensure that each Supporting Information file has a legend listed in the manuscript after the references list.

Reviewers' comments:

Reviewer's Responses to Questions

**Comments to the Author**

1. Is the manuscript technically sound, and do the data support the conclusions?

Reviewer #1: Yes

Reviewer #2: Yes

Reviewer #3: Yes

2. Has the statistical analysis been performed appropriately and rigorously? 

Reviewer #1: Yes

Reviewer #2: N/A

Reviewer #3: Yes

3. Have the authors made all data underlying the findings in their manuscript fully available?

Reviewer #1: Yes

Reviewer #2: Yes

Reviewer #3: Yes

4. Is the manuscript presented in an intelligible fashion and written in standard English?

Reviewer #1: Yes

Reviewer #2: Yes

Reviewer #3: Yes

5. Review Comments to the Author

Reviewer #1: The paper is very well written and clear, the topic is relevant, and the issue is addressed properly.

I suggest a minor revision dealing mainly with the final part. The main body of the paper is very clear in addressing why consideration of both regression parameters in Equation 1 and their variances are relevant to the analysis of intergenerational wealth mobility. However, the final attempt to propose transition matrices as a possible solution (is that it? or have I misinterpreted?) comes across as poorly introduced and not very thorough. It is only in the Conclusion that transition matrices are mentioned as a possible solution. However, they are already widely used in the literature on intergenerational mobility. Moreover, it does not seem to me that they take into account the issue of variances of the parameters, and it is not clear how they take into account the mu parameter.

Therefore, I suggest that, if you believe that transition matrices are a possible solution to the criticisms demonstrated in the rest in the paper, you discuss in more detail why it is the appropriate tool for empirical practice. If, on the other hand, this is not a possible solution for empirical practice, I suggest you better explain why transition matrices are reported in the paper, and making proposals for how to account for both parameters and their variances.

Minor points:

- Note to Figure 2: it seems that the comment to panel A anctually refers to panel B and vice versa.

- The figures would benefit from using different symbols/patterns in addition to colors to be readable even in black and white.

Reviewer #2: This paper investigates the effects of convergence value, μ and the autoregressive coeffi- cient, β to the wealth mobility. In particular the authors show that redistribution (ergodic process) depends on both β and μ. The authors study the wealth mobility using a first- order autoregressive model. This model reflects the statistical relationship between the income (or wealth) of one generation and that of the next generation.

General comments

Comment 1. Authors setting μ = 0 for different circumstances, this eliminates any intrinsic contribution to the average income level that is not influenced by intergenerational mobility. The authors show that for μ = 0 would fall redistribution even for low levels of β.

My comments about this result are:

• Does it make sense to set μ equal to zero? In real-world economic contexts where minimum income/wealth levels are guaranteed by welfare systems or other institu- tions the convergence value si greater than zero. Does these resulst continue to hold even when μ > 0?

Comment 2. Into the Figure 1 (A) the convergence value (red dotted line) is reported as 5, but into the comments associated for the same figure the value of μ is indicated as 0, could be due to an editorial error.

Comment 3. To make the graph (A) in fig.1 more readable the authors can use different color for the two subgroups (f.e. Blue and Red): those starting with wealth 10 and those starting with wealth -10.

Comment 4. For the fig.2 comments seem to me to be reversed compared to the graphs.

The paper is well written and the topic is interesting. I propose for publication after the authors have satisfactorily addressed my questions.

Reviewer #3: Referee Report on Manuscript PONE-D-24-56261: "Is Intergenerational Elasticity (IGE) a Misleading Measure of Wealth Mobility?"

I have reviewed the manuscript "Is Intergenerational Elasticity (IGE) a Misleading Measure of Wealth Mobility?". The study explores critical issues surrounding intergenerational elasticity (IGE) as a measure of wealth mobility, emphasizing the limitations of conventional interpretations. The authors present a theoretical discussion supported by illustrative data simulations and empirical applications.

Although the paper presents an interesting model that identifies the statistical conditions under which the intergenerational elasticity parameter (IGE) should be considered a reliable measure of intergenerational correlations, I don't believe that focusing on the intergenerational wealth elasticity is the best way to criticize the elasticity as the work-horse measure of intergenerational elasticity. Specifically, most of the papers focusing on the intergenerational transmission of wealth choose the rank-rank slope rather than the intergenerational elasticity since the former parameter can also be estimated when the monetary measure of economic status can assume zero or negative values as no log transformation is needed.

Moreover, even in cases when the extent of the intergenerational correlation is highly non-linear across the wealth distribution, there are several ways to account for non-linearities. For instance, one can plot the mean rank of children across percentiles or deciles of the parents' wealth distribution or perform a top decile regression. Local linear kernel regressions can be estimated, too, to avoid a misleading interpretation of intergenerational correlations across the distribution. See Adermon et al. (2018) for further information on how to estimate and consider non-linearities.

I suggest focusing on empirical studies on intergenerational earnings or income mobility to present potential critics of the elasticity, even if the rank-rank slope has been recently very often used in many papers on income mobility, too. (see Chetty (2014), among others).

References

Adrian Adermon, Mikael Lindahl, Daniel Waldenström, Intergenerational Wealth Mobility and the Role of Inheritance: Evidence from Multiple Generations, The Economic Journal, Volume 128, Issue 612, 1 July 2018, Pages F482–F513, https://doi.org/10.1111/ecoj.12535

6. PLOS authors have the option to publish the peer review history of their article (what does this mean?). If published, this will include your full peer review and any attached files.

Reviewer #1: No

Reviewer #2: No

Reviewer #3: No

---

## [Author Response · Author response to Decision Letter 1]

14 Apr 2025

See the PDF file in the attachment. For your convenience, we pasted the text below as well.

Reviewer #1:

\RC The paper is very well written and clear, the topic is relevant, and the issue is addressed properly.

\AR Thank you. We appreciate your feedback.

\RC I suggest a minor revision dealing mainly with the final part. The main body of the paper is very clear in addressing why consideration of both regression parameters in Equation 1 and their variances are relevant to the analysis of intergenerational wealth mobility. However, the final attempt to propose transition matrices as a possible solution (is that it? or have I misinterpreted?) comes across as poorly introduced and not very thorough. It is only in the Conclusion that transition matrices are mentioned as a possible solution. However, they are already widely used in the literature on intergenerational mobility. Moreover, it does not seem to me that they take into account the issue of variances of the parameters, and it is not clear how they take into account the mu parameter.

\RC* Therefore, I suggest that, if you believe that transition matrices are a possible solution to the criticisms demonstrated in the rest in the paper, you discuss in more detail why it is the appropriate tool for empirical practice. If, on the other hand, this is not a possible solution for empirical practice, I suggest you better explain why transition matrices are reported in the paper, and making proposals for how to account for both parameters and their variances.

\AR We agree that the formulation in our previous version of the manuscript could be improved. Transition matrices offer a partial solution to the issues around IGE in that they offer a more fine-grained view of wealth dynamics and transmission. However, the observations we make based on them are dependent on the subdivisions used for the population and they also suffer from information loss. Since we wanted to show the transition matrices as an additional tool rather than an ultimate solution, we added a paragraph after the paragraph discussing the heatmaps (Fig.~8). Moreover, we modified the last paragraph of the conclusion.

\begin{quote}

\edit{While Fig 8 provides insight into the proportion of wealth transmitted across generations, the transmission matrices represented by the heatmaps do not account for the possibility that fathers in the top 20\% wealth bracket may themselves originate from this same bracket of the previous generation [18]. In essence, this visualization is limited in its ability to capture long-term wealth dynamics, as it focuses on static snapshots of wealth transmission between two generations without accounting for broader temporal trends or cumulative effects over multiple generations. Additionally, the observed patterns could vary when subdividing the population into finer wealth categories or incorporating other variables such as surnames [22, 23]. Consequently, Fig 8 should be interpreted as complementary to the earlier analyses, offering a visual perspective on intergenerational wealth mobility rather than a standalone conclusion.}

\edit{

\begin{enumerate}

\item[[18]] Shorrocks AF. Income mobility and the Markov assumption. The Economic Journal. 1976; 86(343):566–578.

\item [[22]] Fields GS, Ok EA. The measurement of income mobility: an introduction to the literature. Handbook of income inequality measurement. 1999; p. 557–598.

\item [[23]] McFarland DD. Intragenerational social mobility as a markov process: Including a time-stationary mark-ovian model that explains observed declines in mobility rates over time. American Sociological Review. 1970; p. 463–476.

\end{enumerate}

}

\section*{Conclusion}

[....]

While we suggested an approach to focus on both parameters and their variances, an alternative method for summarizing analysis results is not proposed. Instead, with data from [11], we demonstrated some visualizations of wealth redistribution across periods with \edit{Kendall rank correlation coefficient and heatmaps. While these can serve as additional tools to investigate the divergences within the population's wealth transmission, further research is needed to develop a practical alternative to the conventional IGE method. Nonetheless, we believe that our approach can already provide valuable insights into wealth mobility, broadening the comparability and representativity of the measure.}

\end{quote}

\subsection{Minor points:}

\RC Note to Figure 2: it seems that the comment to panel A actually refers to panel B and vice versa.

\AR Thank you so much for pointing this out. We changed the order in the figure.\\

\begin{quote}

\centering\includegraphics[width=0.5\textwidth]{img/Fig2. The role of mu and beta.pdf}

\end{quote}

\RC The figures would benefit from using different symbols/patterns in addition to colors to be readable even in black and white.

\AR Thank you for a nice suggestion. We agree that the figures in black and white were not clear. We have made the following changes accordingly:

\begin{quote}

\begin{itemize}

\item \textbf{Fig 1}: We added t=0 and two markers for different initial wealth groups. \\

\centerline{\includegraphics[width=0.5\linewidth]{img/Fig1. Paradoxical relationship.pdf}}

\item \textbf{Fig 4}: We changed the color scale from a bright to a dark color (rather than the previous version with a dark-bright-dark color scheme). Then, we increased the marker size. \\

\centerline{\includegraphics[width=0.5\linewidth]{img/Fig4. The effect of uni- and bimodality.pdf}}

\item \textbf{Fig 5}: We increased the opacity of the lines and added different markers per type of experiment. \\

\centerline{\includegraphics[width=0.5\linewidth]{img/Fig5. Kendall's tau of the cases.pdf}}

\item \textbf{Fig 6}: We added markers for different Nid’s. \\

\centerline{\includegraphics[width=0.5\linewidth]{img/Fig6. Subgroup- and population-wise mu.pdf}}

\item \textbf{Fig 7}: We added markers for different Nid’s. \\

\centerline{\includegraphics[width=0.5\linewidth]{img/Fig7. Subgroup- and population-wise beta.pdf}}

\end{itemize}

\end{quote}

Reviewer #2:

\RC This paper investigates the effects of convergence value, $\mu$ and the autoregressive coefficient, $\beta$ to the wealth mobility. In particular the authors show that redistribution (ergodic process) depends on both $\beta$ and $\mu$. The authors study the wealth mobility using a first-order autoregressive model. This model reflects the statistical relationship between the income (or wealth) of one generation and that of the next generation.

\subsection{General comments}

\RC Comment 1. Authors setting $\mu=0$ for different circumstances, this eliminates any intrinsic contribution to the average income level that is not influenced by intergenerational mobility. The authors show that for $\mu=0$ would fall redistribution even for low levels of $\beta$.

My comments about this result are:

\begin{itemize}

\item Does it make sense to set $\mu$ equal to zero? In real-world economic contexts where minimum income/wealth levels are guaranteed by welfare systems or other institutions, the convergence value is greater than zero. Does these results continue to hold even when $\mu>0$?

\end{itemize}

\AR Thank you for the interesting question! Of course, setting the convergence value as zero in Figs.~4 and 5 is an arbitrary choice we made to achieve a mathematically elegant simulation. While having a zero $\mu$ can make sense when the population’s wealth is centered to its average or normalized to have zero mean (and therefore, the same conclusions hold with the case when $\mu>0$), we see that it may not give an intuitive reflection on the reality. Therefore, we made changes in the simulations for Figs.~4 and 5 to have 5 as a population convergence value.

\begin{quote}

\centering

\includegraphics[width=0.6\textwidth]{img/Fig4. The effect of uni- and bimodality.pdf}\\

\justifying {\bf Fig 4.} {\bf The effect of uni- and bimodality of $\mu$ and $\beta$.} (A) Bimodal $\mu$ case: $\mu_i \sim \mathcal{N}(10, 0.01^2)$ for the half of the population and $\mu_i \sim \mathcal{N}(-10, 0.01^2)$ for the other half and $\beta$ is fixed to 0.45. (B) Unimodal $\mu$ case: $\mu_i \sim \mathcal{N}(\edit{5}, 0.01^2)$ for all with a fixed $\beta=0.45$. (C) Bimodal $\beta$ case: $\beta_i \sim U(0, 0.2)$ for the half of the population and $\beta_i \sim U(0.7, 0.9)$ for the other half, and $\mu$ is fixed to \edit{5}. (D) Unimodal $\beta$ case: $\beta_i \sim U(0.4, 0.5)$ for all with a fixed \edit{$\mu=5$}. (E) Both $\mu$ and $\beta$ have \edit{bimodality}. (F) Both $\mu$ and $\beta$ have \edit{unimodality}. The population size is 100. The expected IGE value at the population level is 0.45 for (C) and (D) and the population-wise convergence value is \edit{5} for all. \\

\centering{\includegraphics[width=0.6\textwidth]{img/Fig5. Kendall's tau of the cases.pdf}}\\

\justifying {\bf Fig 5. Kendall's $\tau$ of the cases with different variances in $\mu$ and $\beta$.} (A) Narrow vs. wide $\mu$ both with a narrowly distributed $\beta$ (A vs. B in Fig~3); (B) Narrow vs. wide $\beta$ both with a narrowly distributed $\mu$ (A vs. C in Fig~3).(C) Bimodal $\mu$ case vs. Unimodal $\mu$ case (D) Bimodal $\beta$ case vs. Unimodal $\beta$ case. The setups are the same as in Fig~4, but 30 replications are made for each scenario. $\tau$ of 0 signifies the state in which the initial wealth's ranking no longer matters. Comparing the results to Fig~3, this reveals that $\tau$ does not directly translate into whether all the population converges to the same wealth level.

\end{quote}

\RC Comment 2. Into the Figure 1 (A) the convergence value (red dotted line) is reported as 5, but into the comments associated for the same figure the value of $\mu$ is indicated as 0, could be due to an editorial error.

\AR Thank you so much for pointing this out. You are correct. We corrected it to 5. Please see the change in our response to the next comment.

\RC Comment 3. To make the graph (A) in fig.1 more readable the authors can use different color for the two subgroups (f.e. Blue and Red): those starting with wealth 10 and those starting with wealth -10.

\AR We changed Fig 1 as you suggested. Just to balance the colors, we also turned the red dashed line into black and used salmon and gray to denote the different starting wealth groups.

\begin{quote}

\centering{\includegraphics[width=0.6\textwidth]{img/Fig1. Paradoxical relationship.pdf}}\\

\justifying {\bf Fig 1. Paradoxical relationship between IGE and wealth dynamics.} Two cases are presented where the log wealth of two groups with contrasting initial wealth evolve over 20-time steps/generations. In (A), the convergence values of the population and the subgroups are different, while they are the same in (B). The population-wise IGE value is 0.3 in (A) with a convergence value of \edit{5}, and the IGE value of 0.8 with a convergence value of 5 in (B). \edit{The subgroup-specific wealth dynamics are gray for those with high initial wealth (\textcolor{gray!30}{\ding{108}}) and red (\textcolor{salmon!50}{\ding{54}}) for those with low initial wealth, and the population-wise convergence value is the black dashed line.} The lower IGE in (A) is preferred to (B) in a conventional interpretation. However, the graphs demonstrate that (B) redistributes wealth across populations of different initial wealth more than (A).

\end{quote}

\RC Comment 4. For the fig.2 comments seem to me to be reversed compared to the graphs.

\AR Thank you again! We changed the order of the plots in Fig.2.

\begin{quote}

\centering \includegraphics[width=0.5\textwidth]{img/Fig2. The role of mu and beta.pdf}

\end{quote}

\RC The paper is well written and the topic is interesting. I propose for publication after the authors have satisfactorily addressed my questions.

\AR Thank you for your feedback.

Reviewer #3:

\RC Referee Report on Manuscript PONE-D-24-56261: "Is Intergenerational Elasticity (IGE) a Misleading Measure of Wealth Mobility?"

\RC* I have reviewed the manuscript "Is Intergenerational Elasticity (IGE) a Misleading Measure of Wealth Mobility?". The study explores critical issues surrounding intergenerational elasticity (IGE) as a measure of wealth mobility, emphasizing the limitations of conventional interpretations. The authors present a theoretical discussion supported by illustrative data simulations and empirical applications.

\RC* Although the paper presents an interesting model that identifies the statistical conditions under which the intergenerational elasticity parameter (IGE) should be considered a reliable measure of intergenerational correlations, I don't believe that focusing on the intergenerational wealth elasticity is the best way to criticize the elasticity as the work-horse measure of intergenerational elasticity. Specifically, most of the papers focusing on the intergenerational transmission of wealth choose the rank-rank slope rather than the intergenerational elasticity since the former parameter can also be estimated when the monetary measure of economic status can assume zero or negative values as no log transformation is needed.

\AR Thank you for raising this crucial question. As you correctly point out, the rank-rank slope is a widespread and related measure of wealth mobility and, therefore, merits a discussion in this paper. We have added a new section that introduces the rank-rank measure and points out that it suffers from the same shortcomings as IGE.

\begin{quote}

\section*{\edit{Rank-Rank slope and accounting for the zero-wealth problem}}

\edit{In our experiments, non-zero wealth is assumed for all the individuals given the conventional IGE model with the log-transformed wealth. However, one may encounter a situation where zero-wealth (or zero income or earnings) cases must be accounted for. In those cases, one may consider using the rank-rank coefficient rather than the IGE to handle the issue with the zero-wealth cases as Adermon et al. (2018) suggested. The rank-rank coefficient is given by:}

\edit{

\begin{equation*}

r_{it+1} = a + b r_{it} + \epsilon_{it} ,

\end{equation*}

}

\edit{where $r$ is the ranking of the $i^\text{th}$ subgroup at time $t$, $a = m(1-b)$ and $\epsilon_{it}$ is a white noise. When $|b|<1$,}

\edit{

\begin{equation*}

r_{it} = b^t r_{i0} + (1-b^t)m ,

\end{equation*}

}

\edit{which shows that in limit, $r_{it}$ converges to $m$. With or without the log transformation of the variable, the AR(1) structure assumes ergodicity. Therefore, when using such a structure, it will be crucial to consider both the convergence value and the slope estimate and their variances.}

\end{quote}

\RC Moreover, even in cases when the extent of the intergenerational correlation is highly non-linear across the wealth distribution, there are several ways to account for non-linearities. For instance, one can plot the mean rank of children across percentiles or deciles of the parents' wealth distribution or perform a top decile regression. Local linear kernel regressions can be estimated, too, to avoid a misleading interpretation of intergenerational correlations across the distribution. See Adermon et al. (2018) for further information on how to estimate and consider non-linearities.

\AR Indeed, when analyzing empirical data, there are challenges of non-linearity. In Adermon et al. (2018), the authors used top decile regressions to account for the nonlinearity observed in the top decile. Our paper also aims to highlight the possible nonlinearity in data, specifically recognizing that not every decile/subgroup transfers their wealth to the next generation in the same manner. As you suggested, we see that different approaches could be used to account for this problem. However, we would like to focus on pointing out the issues with the conventional use of IGE that lacks comparability and representativity.

\RC I suggest focusing on empirical studies on intergenerational earnings or income mobility to present potential critics of the elasticity, even if the rank-rank slope has been recently very often used in ma

---

## [Editor Report · Decision Letter 1]

23 Apr 2025

Is Intergenerational Elasticity (IGE) a Misleading Measure of Wealth Mobility?

PONE-D-24-56261R1

Dear Dr. Kim,

We’re pleased to inform you that your manuscript has been judged scientifically suitable for publication and will be formally accepted for publication once it meets all outstanding technical requirements.

Kind regards,

Marco Maria Sorge, PhD

Academic Editor

PLOS ONE
---

## [Editor Report · Acceptance letter]

PONE-D-24-56261R1

PLOS ONE

Dear Dr. Kim,

I'm pleased to inform you that your manuscript has been deemed suitable for publication in PLOS ONE. Congratulations! Your manuscript is now being handed over to our production team.

Kind regards,

on behalf of

Professor Marco Maria Sorge

Academic Editor

PLOS ONE